# Identification of B-Cell Linear Epitopes in the Nucleocapsid (N) Protein B-Cell Linear Epitopes Conserved among the Main SARS-CoV-2 Variants

**DOI:** 10.3390/v15040923

**Published:** 2023-04-06

**Authors:** Rodrigo N. Rodrigues-da-Silva, Fernando P. Conte, Gustavo da Silva, Ana L. Carneiro-Alencar, Paula R. Gomes, Sergio N. Kuriyama, Antonio A. F. Neto, Josué C. Lima-Junior

**Affiliations:** 1Laboratory of Immunological Technology, Institute of Technology in Immunobiologicals, FIOCRUZ, Rio de Janeiro 21040-900, Brazil; gustavosilva@aluno.fiocruz.br (G.d.S.); anaalencar@aluno.fiocruz.br (A.L.C.-A.); 2Eukaryotic Pilot Laboratory, Institute of Technology in Immunobiologicals, FIOCRUZ, Rio de Janeiro 21040-900, Brazil; fernando.conte@bio.fiocruz.br; 3Laboratory of Immunoparasitology, Oswaldo Cruz Institute, FIOCRUZ, Rio de Janeiro 21040-900, Brazil; 4Getulio Vargas State Hospital, Rio de Janeiro 21070-061, Brazil; 5SENAI Innovation Institute for Green Chemistry, Rio de Janeiro 20271-030, Brazil

**Keywords:** COVID-19, immunoinformatic, antigen test, SARS-CoV-2 variants

## Abstract

The Nucleocapsid (N) protein is highlighted as the main target for COVID-19 diagnosis by antigen detection due to its abundance in circulation early during infection. However, the effects of the described mutations in the N protein epitopes and the efficacy of antigen testing across SARS-CoV-2 variants remain controversial and poorly understood. Here, we used immunoinformatics to identify five epitopes in the SARS-CoV-2 N protein (N_(34–48)_, N_(89–104)_, N_(185–197)_, N_(277–287)_, and N_(378–390)_) and validate their reactivity against samples from COVID-19 convalescent patients. All identified epitopes are fully conserved in the main SARS-CoV-2 variants and highly conserved with SARS-CoV. Moreover, the epitopes N_(185–197)_ and N_(277–287)_ are highly conserved with MERS-CoV, while the epitopes N_(34–48)_, N_(89–104)_, N_(277–287)_, and N_(378–390)_ are lowly conserved with common cold coronaviruses (229E, NL63, OC43, HKU1). These data are in accordance with the observed conservation of amino acids recognized by the antibodies 7R98, 7N0R, and 7CR5, which are conserved in the SARS-CoV-2 variants, SARS-CoV and MERS-CoV but lowly conserved in common cold coronaviruses. Therefore, we support the antigen tests as a scalable solution for the population-level diagnosis of SARS-CoV-2, but we highlight the need to verify the cross-reactivity of these tests against the common cold coronaviruses.

## 1. Introduction

Severe acute respiratory syndrome coronavirus 2 (SARS-CoV-2) is responsible for the current worldwide outbreak, which less than four months after the first case in China [1] was declared by World Health Organization (WHO) as a pandemic [2]. Until now, the coronavirus disease 2019 (COVID-19) affected more than 663 million people and resulted in about 7 million deaths throughout the globe [3]. Currently, despite the use of vaccines to tackle the spread of the virus and minimize the associated morbidity and mortality [4], more than 80,000 cases and about 500 deaths still are daily reported around the world [3]. In this scenario, with about 37 million COVID-19 confirmed cases and more than 698,000 deaths, Brazil currently figures as the second country with more deaths and as the sixth in COVID-19 cases around the world [3].

SARS-CoV-2, like all RNA viruses, is susceptible to rapidly accumulating mutations in its genome [5] leading to changes that can allow the virus to evade the immune response or affect its ability to spread or cause disease, resulting in the rise of lineages that are classified by the Center of Diseases Control and Prevention (CDC) as “variants of interest (VOIs)”, or “variants of concern (VOCs)”, according to its characteristics that impact public health [6]. In this context, the Omicron B.1.1.529 variant (and multiple subvariants) is more transmissible than previous variants and now accounts for the majority of infections [7]. Nowadays, there are several SARS-CoV-2 variants (such as Alpha, Beta, Delta, Epsilon, Eta, Iota, Kappa, Mu, and Omicron); however, only in the Omicron variant are there more than 600 Pango lineages [8]. The steep increase in the number of SARS-CoV-2 variants is a critical obstacle to COVID-19 eradication and reinforces the need for constant and effective surveillance. In addition to SARS-CoV-2 and its variants, there are four common cold coronaviruses (229E, NL63, OC43, and HKU1) that circulate in the human population, usually causing mild respiratory symptoms similar to the common cold, such as cough, runny nose, and sore throat, that should be investigated about cross-reactivity in COVID-19 diagnosis [9,10,11,12].

Regarding diagnostic tests for the surveillance of COVID-19 cases, despite RT-PCR remaining the definitive method to diagnose infection, rapid antigen tests offer a valuable way to offer mass testing, since they can be performed without the need for specialized training or resources [13,14,15]. Among the SARS-CoV-2 structural proteins, while the Spike (S) protein is the immunodominant antigen and the main target for neutralizing antibodies and vaccines, the nucleocapsid (N) protein is highlighted as the main target for diagnostic by antigen detection due to its abundance in circulation early during infection [16,17]. Remarkably, most of the mutations described in SARS-CoV-2 variants are in the S protein, while the N protein seems to be highly conserved among variants and lineages [5,8,18,19]. However, the effects of the described mutations in the N protein epitopes and the efficacy of the antigen test remain unexplored. Considering this, this study aims to identify in silico and experimentally validate B-cell linear epitopes in the SARS-CoV-2 N protein and to evaluate their conservation across main SARS-CoV-2 variants and lineages.

## 2. Materials and Methods

### 2.1. Sequences Data and 3D Structures

To predict possible antigenic properties and select potential B-cell epitopes, the sequence of SARS-CoV-2 N protein (UniProt ID: P0DTC9) was used. The complete structure of the N protein was modeled using the Robetta server (http://new.robetta.org/, accessed on 10 September 2020) [20,21] based on the full-length amino acid sequence of the protein. This server is continually evaluated through CAMEO (Continuous Automated Model Evaluation) and generates five models analyzed by MolProbity (molprobity.biochem.duke.edu; accessed on 20 September 2020), which is a widely used system of model validation for protein structures. The best predictive model was selected and used in further analysis.

### 2.2. In Silico Prediction of Linear B-Cell Epitopes

We used a combination of web-based tools for B-cell epitope prediction: the Immune Epitope Database (IEDB) [22] and ABCpred [23] servers.

The IEDB (http://www.iedb.org/, accessed on 10 March 2020) is a freely available resource funded by NIAID. This server catalogs experimental data on antibody and T cell epitopes studied in humans, non-human primates, and other animal species in the context of infectious disease, allergy, autoimmunity and transplantation. The IEDB also hosts tools to assist in the prediction and analysis of epitopes. In this study, we used the ElliPro [24], Bepipred 1 [25] and EMINI Surface Accessibility [26] modules on the IEDB server with default settings to define B-cell linear epitopes exposed on the protein surface. We also used the ABCpred server [23] to refine our prediction using an artificial neural network (ann) method. All algorithms were accessed on 10 March 2020. Finally, predicted sequences with more than 9 mers and that were predicted by at least three of the algorithms were defined as linear B-cell epitopes.

### 2.3. Prediction of Antigenicity

To exclude non-antigenic sequences, the predicted linear B-cell epitopes were evaluated by the VaxiJen server (accessed in 12 March 2020), which is the first server for alignment-independent prediction of protective antigens. Its algorithm was developed to allow antigen classification solely based on the physicochemical properties of proteins without recourse to sequence alignment. Viral datasets were selected to derive models for the prediction of whole protein antigenicity, showing prediction accuracy from 70 to 89% [27,28,29]. Using the default threshold (0.5), all sequences predicted as non-antigenic were excluded from the study.

### 2.4. Peptide Synthesis

The best predicted epitopes were synthesized by fluorenylmethoxycarbonyl (F-moc) solid-phase chemistry [30,31] (GenOne Biotechnologies, Rio de Janeiro, Brazil). Analytical chromatography of the peptide demonstrated a purity of >95%, and mass spectrometric analysis also indicated estimated masses corresponding to the molecular masses of predicted peptides.

### 2.5. Patients and Samples

Samples from convalescent COVID-19 donors: Twenty individuals (12 women and 8 men), with ages ranging from 25 to 51 years (mean age: 35.8 ± 6.7 years) and confirmed SARS-CoV-2 infections who had been tested using real-time RT-PCR for viral infections or who had tested positive in the serological assay for COVID-19, were invited to enroll in the study. The serum samples were collected only in the convalescent phase. After recovering from COVID-19, convalescent donors were screened for symptoms and had to be symptom-free and approximately 3 weeks out from symptom onset at the time of the blood draw. Asymptomatic individuals, who had had contact with infected patients and were positively tested by RT-PCR but who had not presented symptoms for at least 21 days post-diagnosis, were also invited to enroll in the study. All donors voluntarily gave their informed consent before being enrolled in the study. Individuals did not receive compensation for their participation.

Healthy unexposed donors: A total of 20 samples (13 women and 7 men), from blood donors, in Brazilian blood centers between the years 2010 and 2018 with ages ranging from 20 to 56 years (36.7 ± 10.4 years) were randomly selected from the serum biobank for the development of diagnostic tests of the Institute of Technology in Immunobiologicals. These samples were considered to be from unexposed controls, given that SARS-CoV-2 emerged as a novel pathogen in late 2019, more than one year after the collection of any of these samples.

Peripheral blood samples were collected by venipuncture in EDTA tubes. After centrifugation (350× *g*, 10 min), the plasma was collected and stored at −30 °C.

Written informed consent was obtained from all COVID-19 donors, and the study was reviewed and approved by the Oswaldo Cruz Foundation Ethical Committee and the National Ethical Committee of Brazil (CEP-FIOCRUZ CAAE 31368620.0.0000.5262).

### 2.6. Antibody Assays

Plasma samples from donors were screened for the presence of naturally acquired antibodies against the S-ECD and S-RBD recombinant proteins and synthetic peptide, predicted as linear B-cell epitopes in SARS-CoV-2 N protein, by enzyme-linked immunosorbent assay (ELISA) essentially as previously described [32,33,34]. Briefly, MaxiSorp 96-well plates (Nunc, Rochester, NY, USA) were coated with PBS containing 5 µg/mL of recombinant protein or 20 µg/mL of a peptide. After overnight incubation at 4 °C, plates were washed with PBS and blocked with PBS-Tween containing 5% skim milk (PBS-Tween-M) for 1 h at 37 °C. Individual plasma samples diluted 1:100 on PBS-Tween-M were added in duplicate wells, and the plates were incubated at 37 °C for 2 h. After three washes with PBS-Tween, bound antibodies were detected with peroxidase-conjugated goat anti-human IgM (Sigma, St. Louis, MO, USA, cat number A 6907) or peroxidase-conjugated goat anti-human IgG (Southern, AL, cat number 2040-05) followed by the addition of 3,3′,5,5′-tetramethylbenzidine (Sigma St. Louis, MO, cat number N301). Optical density was measured at 450 nm using a SpectraMax microplate spectrophotometer (Molecular Devices, Sunnyvale, CA, USA). The results for total IgM and IgG were expressed as reactivity indexes (RIs), which were calculated by the ratio between the mean optical density of an individual’s tested sample and the mean optical density samples of 20 unexposed individuals plus 2.5 standard deviations. Subjects were classified as responders to an antigen if the RI of IgM or IgG were higher than 1.

### 2.7. Conservancy Analysis of the Selected Epitopes across SARS-CoV-2 Variants and Other Human Coronaviruses

To evaluate the conservancy of naturally antigenic epitopes among SARS-CoV-2 lineages and variants around the world, we listed the main mutations reported in GISAID-Lineage comparison (https://gisaid.org/lineage-comparison/, accessed on 20 February 2023) [35]. GISAID (Global Initiative on Sharing Avian Influenza Data) is a global science initiative that provides open access to the genomic data of influenza viruses [36] and the coronavirus responsible for the COVID-19 pandemic [37,38]. In this study, we looked for mutations described in the SARS-CoV-2 N-protein from previously Circulating Variants of Concern (Alpha, Beta, Gamma, and Delta) and from Current Variants of Concern (Omicron, Omicron—XBB.1.5.X, Omicron—CH.1.1.X, Omicron—BA.2.75.X, Omicron—BA.5, Omicron BQ.1.X, BQ1.1, and XBB.1), according to GISAID-Lineage comparison. Moreover, we also listed the mutations observed on the top 5 growing lineages in the world on 27 February 2023 (XBB.1.5, CH.1.1.1, BM.4.1, BM.4.1.1, XBB.1.9) and the top five most counted lineages in Brazil until February 2023 (B.1.617.2, P.1.1, BA.2, AY.43, AY.34.1), according to the NCBI-SARS-CoV-2 Variants Overview [8]. The main mutations observed in the SARS-CoV-2 N-protein and their frequencies across SARS-CoV-2 variants are listed in Appendix A.

Furthermore, the conservancy of SARS-CoV-2 N-protein B-cell epitopes was compared with the N-proteins from other human coronavirus: SARS-CoV (Uniprot ID: P59595), MERS-CoV (ID: K9N4V7), HCoV-229E (ID: P15130), HCoV-NL63 (ID: Q6Q1R8), HCoV-OC43 (ID: P33469), and HCoV-HKU1 (ID: Q5MQC6), using the IEDB server (https://www.iedb.org/conservancy/ accessed on 10 March 2021), with a sequence identity threshold at ‘>20′.

### 2.8. In Silico Conservancy Analysis of Amino Acid Residues Recognized by Antibodies

To investigate the cross-reactivity of described antibodies against the N protein across the SARS-CoV-2 variants and other coronaviruses, we selected the crystallographic structures of two nanobodies (PDB: 7R98 and 7N0R) and one antibody (PDB: 7CR5). Using the LigPlus, we listed the main SARS-CoV-2 N protein amino acids recognized by antibody/nanobody and compared the conservation of these amino acids between SARS-CoV-2 variants and other coronaviruses, which were aligned by MUSCLE using Meg Align Pro on DNASTAR Lasergene software.

### 2.9. Statistical Analysis

All statistical analysis was carried out using Prism 5.0 for Windows (GraphPad Software, Inc.). The one-sample Kolmogorov–Smirnoff test was used to determine whether a variable was normally distributed. The Wilcoxon matched pairs test or the paired T-test was used to compare the reactivity indexes of synthetic peptides and recombinant proteins. Differences in the frequency of responder of IgM and/or IgG responders to recombinant proteins were evaluated by chi-square test (χ^2^).

## 3. Results

### 3.1. Prediction of Serological Targets: Linear B-Cell Epitopes in N Protein

Aiming to identify potential serological targets in the SARS-CoV-2 N protein, sequences that were fully or partially predicted as B-cell linear epitopes by at least two of the used prediction algorithms (Bepipred 1.0, ABCpred, EMINI Surface Accessibility, and Ellipro) and also predicted as antigenic (Vaxijen score > 0.5) were screened as predicted epitopes on SARS-CoV-2 N protein. As summarized in Table 1, eleven sequences were predicted as linear B-cell epitopes; however, only five sequences (N_(34–48)_; N_(89–104)_; N_(185–197)_; N_(277–287)_ and N_(378–390)_) were also predicted as antigenic, presenting a Vaxijen score > 0.5 and were synthesized as peptides to be tested against the plasma of COVID-19 convalescent donors. Corroborating this result, the predicted epitopes were exposed in the N-protein 3D structure, as demonstrated in Figure 1.

### 3.2. Profile of Convalescent COVID-19 Donors

To experimentally validate B-cell linear epitopes on SARS-CoV-2 N proteins, plasma samples of 20 convalescent COVID-19 donors were obtained in the early months of the pandemic COVID-19 in Brazil: in 2020 July and June. All studied individuals were from the state of Rio de Janeiro in Brazil, where more than 130,000 cases and about 12,000 deaths were reported until early 2020 July. All convalescent donors had recovered from COVID-19 and were screened for symptoms before scheduling blood draws. They had to be symptom-free and approximately 3 weeks out from symptom onset at the time of the blood draw. Regarding the diagnosis of SARS-CoV-2 infection, 60% of donors were positive diagnosed only by RT-PCR to SARS-CoV-2, 30% were positive only by commercial serological assay and 10% of tested individuals were positive by both methods. About the clinical spectral, 80% of donors experienced mild illness and reported fatigue, fever, headache, and cough as the most common symptoms; 10% of donors presented asymptomatic cases, diagnosed by RT-PCR and serological methods, and persisted without symptoms for 25 days between the molecular diagnose and blood draw, while two donors presented complications (thrombosis and bacterial pneumonia); both already recovered at the moment of blood draw. The characteristics of the studied individuals are summarized in Table 2.

### 3.3. Evaluation of Natural Immunogenicity of Predicted B-Cell Epitopes

Predicted epitopes (N_(34–48)_, N_(89–104)_, N_(185–197)_, N_(277–287)_, and N_(378–390)_) were synthesized as peptides and tested against samples of COVID-19 convalescent donors and healthy unexposed donors. First, as illustrated in Figure 2, we confirmed that the four predicted epitopes were naturally antigenic in SARS-CoV-2 infections since they were recognized by COVID-19 convalescent donors at different levels and not by healthy individuals unexposed to COVID-19. As illustrated in Figure 2, regarding the specific immune response against each epitope, we observed a statistically similar overall frequency of responders against epitopes that ranged from 40% to 75%. The frequencies of IgM responders were higher against the epitopes N_(34–48)_ (50%) and N_(277–287)_ (55%) than against the peptides N_(89–104)_ (0%, *p* = 0.0004 and *p* = 0.0001, respectively), N_(185–197)_ (10%) and N_(378–390)_ (10%) (*p* = 0.0137 and *p* = 0.0057, respectively). In addition, we observed higher frequencies of IgG responders against the epitopes N_(185–197)_ (40%) and N_(277–287)_ (40%) than against the N_(34–48)_ (5%, *p* = 0.0197). Comparing the frequencies of IgG and IgM responders against each epitope, we observed the prevalence of IgM responders against the epitope N_(34–48)_ (*p* = 0.0033) and the prevalence of IgG responders against N_(89–104)_ (*p* = 0.0471) (Figure 3a).

Regarding the magnitude of antibody response, we observed no statistical differences between the IgM levels against N_(34–48)_ (mean: 1.33 ± 0.31) and N_(277–287)_ (mean: 1.71 ± 0.61) (*p* = 0.0986), which are the two epitopes with higher frequencies of responders of IgM responders. In the same way, the IgG levels were similar against the most prevalent epitopes: N_(89–104)_ (mean: 2.28 ± 1.26), N_(185–197)_ (mean: 3.76 ± 3.16), N_(277–287)_ (mean: 4.22 ± 4.10), and N_(378–390)_ (mean: 3.83 ± 3.09) (Figure 3b). Remarkably, 80% of COVID-19 convalescent donors presented antibodies against at least one of the identified epitopes, 15% of studied individuals presented antibodies against only one epitope (10% against N_(277–287)_, and 5% against N_(34–48)_), 15% were responsive against two epitopes (N_(277–287)_ and _(34–48)_), 20% were reactive against three epitopes simultaneously (10% against N_(185–197)_, N_(277–287)_ and _(378–390)_); 5% against N_(89–104),_ N_(185–197)_, and N_(277–287)_; and 5% against N_(34–48),_ N_(185–197)_, and N_(277–287)_), 15% were reactive against four epitopes (10% against N_(34–48),_ N_(185–197)_, N_(277–287)_, and N_(378–390)_; and 5% against N_(34–48)_, N_(185–197)_, N_(277–287)_, and N_(378–390)_) and 15% of convalescent donors presented antibodies against all identified epitopes.

### 3.4. Analysis of Epitope Conservation across SARS-CoV-2 Variants and Lineages

As illustrated in Appendix A, there are 16 key mutations (D3L, Q9L, P13L, Del31/33, D63G, P80R, E136D, R203M, R203K, G204R, T205I, G215C, L230F, S235F, D377Y, S413R) reported across 12 variants of concern (Alpha, Beta, Gamma, Delta, Omicron, Omicron—XBB.1.5.X, Omicron—CH.1.1.X, Omicron—BA.2.75.X, Omicron—BA.5, Omicron—BQ.1.X, BQ1.1, and XBB.1) and across the five growing lineages in the world (XBB.1.5, CH.1.1.1, BM.4.1, BM.4.1.1, XBB.1.9) and the top five most reported lineages in Brazil (B.1.617.2, P.1.1, BA.2, AY.43, AY.34.1). Remarkably, all the five identified epitopes were completely conserved across studied variants, since there are no mutations detected inside their sequences. As shown in Figure 4, there are mutations located close to the epitopes N_(34–48)_, N_(89–104)_, N_(185–197)_, and N_(378–390)_; however, these changes do not seem to affect the identified epitopes, resulting in no alterations in antigenicity or exposition. Moreover, until now, there are no mutations described in the main SARS-CoV-2 variants near epitope N_(277–287)_.

### 3.5. Analysis of Epitope Conservation across Other Human Coronaviruses

After verifying the conservancy of identified epitopes among SARS-CoV-2 isolates, we compare its conservancies among other human coronaviruses (SARS-CoV, MERS-CoV, HCoV-229E, HCoV-NL63, HCoV-OC43, and HCoV-HKU1). Firstly, the epitopes N_(34–48)_, and N_(89–104)_ were highly conserved (> 80%) with SARS-CoV but lowly conserved with MERS-CoV and common cold coronaviruses. In the same way, the epitope N_(277–287)_ was fully conserved with SARS-CoV and moderately conserved among MERS-CoV (Conservancy = 73%) and HCoV-HKU1 (conservancy = 55%), while the epitope N_(185–197)_ was moderately conserved (50% < conservancy < 80%) among ‘common cold’ coronaviruses (HCoV-NL63, HCoV-OC43, and HCoV-HKU1) and also highly conserved among epidemic coronaviruses (SARS-CoV and MERS-CoV) and the epitope N_(378–390)_ was the less conserved epitope, presenting 77% of identity with SARS-CoV and less than 40% of identity with MERS and common cold coronaviruses (Table 3).

### 3.6. Evaluation of Antibody Cross-Reaction against SARS-CoV-2 Variants and Other Coronaviruses

To investigate the cross-reactivity of the described antibodies across the SARS-CoV-2 variants and other coronaviruses, we selected the crystallographic structures of two nanobodies (PDB: 7R98 and 7N0R) and one antibody (PDB: 7CR5). Using the LigPlus, we listed the main SARS-CoV-2 N protein amino acids recognized by antibody/nanobody and compared the conservation of these amino acids between SARS-CoV-2 variants and other coronaviruses.

As shown in Table 4, all amino acids recognized by an antibody were completely conserved across SARS-CoV-2 variants once these amino acids were not among the 16 key mutations related to main variants (D3L, Q9L, P13L, Del31/33, D63G, P80R, E136D, R203M, R203K, G204R, T205I, G215C, L230F, S235F, D377Y, S413R). Regarding the cross-reactivity with other human coronaviruses, we observed different levels of conservation across the human coronaviruses (SARS, MERS, 229E, NL63, OC43, and HKU1). Firstly, the recognized amino acids were completely conserved (100% of identity) with SARS-CoV and highly conserved with MERS-CoV once the residues recognized by the nanobodies 7N0R and 7R98 presented more than 75% of identity and residues recognized by the antibody 7CR5 presented more than 50% of identity with the SARS-CoV-2 N protein.

On the other side, when compared to common cold coronaviruses N proteins, the recognized residues presented lower conservation, which ranged from 0% to 75% of identity. Interestingly, among the three evaluated antibodies, while the residues recognized by nanobody 7N0R and antibody 7CR5 presented a low conservation degree (ranging from 0% to 33% of identity), the residues recognized by the nanobody 7R98 were conserved in coronaviruses 229E (50% of identity) and NL63 (75% of identity), and they were lowly conserved in coronaviruses OC53 and HKU1 (25% of identity) (Table 4).

## 4. Discussion

Despite the global use of vaccines to control the disease, with a drastic reduction in cases and deaths by COVID-19, the continuous rise of new SARS-CoV-2 variants makes constant and large surveillance essential to avoiding new waves and the spread of infection. In this scenario, due to being highly immunogenic and abundantly present in blood and saliva during early asymptomatic and symptomatic SARS-CoV-2 infection [39,40], the N protein is highlighted as the major target for antigen detection, which is a valuable and inexpensive strategy for COVID-19 epidemiological surveillance [13]. However, studies investigating the sensitivity of antigen tests for SARS-CoV-2 variants [41,42,43] raise the question of the truthful efficacy of antigen detection tests against current and future variants. Therefore, this study aimed to identify B-cell linear epitopes in the SARS-CoV-2 N protein and to investigate their conservation across the variants of concern and variants of interest.

Even before COVID-19 became a pandemic, in silico studies were viewed as an innovative, fast, and necessary approach to tackling the disease [44,45,46]. However, despite epitope prediction being a strategy used in several studies against COVID-19 [47,48,49,50], the constant investigation of epitopes and their cross-reactivities across virus variants in different populations is indispensable to confirm the efficacy of vaccines, therapies, and diagnostic tests. Here, we predicted five B-cell linear epitopes in the SARS-CoV-2 N protein (N_(34–48)_, N_(89–104)_, N_(185–197)_, N_(277–287)_, and N_(378–390)_) and confirmed their natural immunogenicity based on their recognition by samples from Brazilian COVID-19 convalescent donors. These data are in agreement with previous studies of our group in which we experimentally validated more than 80% of predicted epitopes from viruses [51,52], bacteria [53,54], and protozoa [55,56]. Moreover, our data corroborate previous studies that suggested similar regions as targets of antibodies [4,50,57,58].

Regarding the recognition of identified epitopes, 80% of studied individuals recognized at least one of these, with a prevalence of IgM response against N_(34–48)_, a prevalence of IgG response against N_(89–104)_, and similar frequencies of IgM and IgG reactivity against the epitopes N_(185–197)_, N_(277–287)_, and N_(378–390)_. Interestingly, these data corroborate the IgG prevalence observed against the peptide 96–100 in the study of Wang and collaborators but disagree with the prevalence of IgG response against peptides 386–390 and 366–400, which was observed in the same study [59]. These differences between studies can be related to differences between Brazilian and Chinese populations’ genetics or in the stage of infection since samples were from patients in the convalescent phase in our study and early-stage patients in the Wang et al. study [59]. Notably, the number of individuals studied is a limitation of our study, but we believe it is sufficient to validate the natural immunogenicity of epitopes. In support of this point, the aforementioned study by Wang et al. used samples from only 10 patients to identify peptides recognized by antibodies [59]. However, we believe that a larger number of samples is needed to evaluate the true sensitivity and specificity of these epitopes in serological assays.

Concerning the use of antigen detection tests in COVID-19 massive surveillance, we investigated the conservation of predicted epitopes across the main variants. Remarkably, we did not observe mutations inside the identified epitopes, suggesting that the identified epitope should be recognized by the same antibodies in the main SARS-CoV-2 variants. These data are inconsistent with the study by Kumar et al., who mapped mutations in predicted B cell linear epitopes in the SARS-CoV-2 N protein [4]. From our point of view, this discrepancy could be related to the selection of the variants studied, since we selected the top growing lineages in the world and the most common variants in Brazil, while Kumar’s study used variations present in the N-protein among 831 Indian isolates of SARS-CoV-2 [4], resulting in a local observation of the mutations.

Leuzinger et al. performed a study comparing different immunoassays and showed that the full-length N protein was cross-recognized by pre-existing antibodies, which were produced during previous exposures to other human coronaviruses [60]. Based on this finding, we investigated the conservation of the identified epitopes in other human coronaviruses (SARS, MERS, 229E, NL63, OC43, and HKU1) and found that all identified epitopes are highly conserved in the SARS-CoV N protein, and epitopes N_(185–197)_ and N_(277–287)_ share more than 70% identity with the MERS-CoV N protein. Notably, despite the high conservation compared to SARS and MERS, the identified epitopes share low conservation (identities ranging from 23% to 61%) with common cold coronaviruses (229E, NL63, OC43, and HKU1), suggesting that these epitopes are mainly recognized by antibodies specifics to SARS-CoV-2′s N protein with low cross-reactivity against the most common coronaviruses. These data corroborate studies that suggested the use of N domains with deleted conserved regions as a target for specific antibody-based assays for COVID-19 [60,61]. Furthermore, our study corroborates the study by Wen et al., reporting that monoclonal antibodies to the SARS-CoV-2 nucleocapsid protein cross-react with their counterparts of SARS-CoV but not other human betacoronaviruses [62], and it suggests that knowing the epitope conservation shared among human coronaviruses is a critical step in predicting the cross-reactivity of a protein as an antigen to the serologic test, as the epitope conservation does not necessarily reflect the conservation shared in the whole protein [63,64].

Considering that the N protein is the main target of antigen detection tests and that the sensitivity of the test depends on the specificity and affinity of the antibodies used to detect it, we evaluated the conservation of amino acids recognized by three anti-antibodies to SARS-CoV-2 N protein among the major SARS-CoV-2 variants and other human coronaviruses. Remarkably, the amino acid residues recognized by the three antibodies studied were not located in key-point mutations positions, suggesting that these antibodies can recognize the N protein of the main SARS-CoV-2 variants and corroborating the hypothesis that the decrease in sensitivity of antigen tests can be related to other causes, such as a lower viral load of some variants [43]. Furthermore, our data suggest that these antibodies to the SARS-CoV-2 N protein can cross-recognize the protein of SARS and MERS, corroborating previous studies that demonstrated that antibodies against N protein from SARS and MERS can cross-react with the SARS-CoV-2 protein [64,65,66], and reinforcing that caution should be used while interpreting assay results when the full-length recombinant N protein of SARS-CoV-2 is used as a reagent for the diagnosis of SARS-CoV-2 infections in humans.

Conversely, despite the cross-reactivity with SARS and MERS, our data support the use of antigen testing as a tool for massive epidemiological surveillance of COVID-19. Considering that both coronaviruses, SARS and MERS, are not usually reported worldwide, the common cold coronaviruses (229E, NL63, OC43, and HKU1) should be the main concern for antigen test cross-reactivity. In this context, regarding the potential cross-reactivity with N-protein from common cold coronaviruses, only one studied antibody seems to recognize amino acid conserved in coronaviruses 229E and NL63. Therefore, despite our data showing that most of the identified epitopes are poorly conserved across common cold coronaviruses, there are epitopes identified and amino acids recognized by antibodies that are highly conserved across these viruses. Several studies have reported differences in the sensitivity of antigen tests to detect SARS-CoV-2 infection [67,68,69,70], which may be related to patient characteristics such as days of symptoms, virus variant characteristics such as viral load, or characteristics of the antibodies used in the test, such as affinity and specificity. These data highlight the need to evaluate the cross-reactivity of antigen tests against common cold coronaviruses to prove their specificity.

Concluding, this study gives a comprehensive view of B-cell epitopes of SARS-CoV-2 N protein and their conservation across the main SARS-CoV-2 variants and other coronaviruses. Despite studies that have demonstrated that mutations in the Spike protein of SARS-CoV-2 variants may lead to decreased sensitivity to neutralizing antibodies [71,72,73], it is evident from our study that the main mutations described in major SARS-CoV-2 variants are not inserted in identified epitopes, neither in amino acids recognized by the evaluated antibodies. In this context, our study supports the use of antigen testing as a scalable solution for population-level diagnosis of SARS-CoV-2; however, we highlight the need to verify the cross-reactivity of these tests against the N protein of common cold coronaviruses, which are described worldwide.

## 5. Conclusions

Our study validated experimentally five predicted B-cell linear epitopes as naturally immunogenic by their reactivity against the serum samples from COVID-19 convalescent patients. Remarkably, these epitopes were conserved across the major SARS-CoV-2 variants, suggesting that antigen tests based on antibodies specific to these epitopes could recognize the N protein of these variants. We also examined the conservation of these epitopes and the amino acid residues recognized by the three antibodies across other human coronaviruses and showed that there is an increased likelihood of cross-reaction with SARS and MERS coronaviruses and a decreased likelihood of cross-reaction with common cold coronaviruses, but these cross-reactions should be verified by antigen test developers.

## Figures and Tables

**Figure 1 viruses-15-00923-f001:**
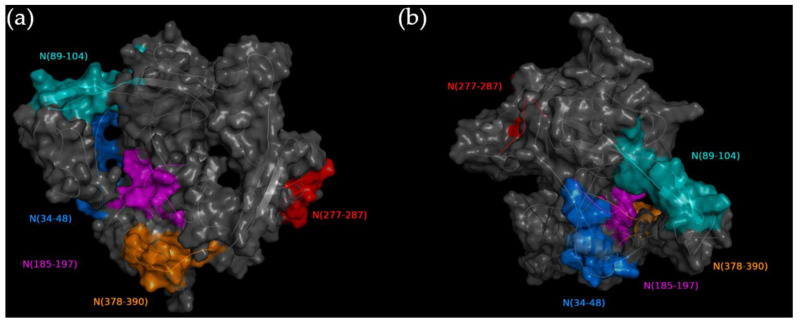
Location of predicted epitopes in SARS-CoV-2 N protein 3D structure. The protein chain is indicated by a gray cartoon and transparent surface. The locations of epitopes N_(34–48)_, N_(89–104)_, N_(185–197)_, N_(277–287),_ and N_(378–390)_ were indicated by colors blue, teal, purple, red, and orange, respectively. In the cartoon, round helices, flat sheets, and smooth loops are applied to allow better visualization of the predicted structure. Different rotations of the protein are shown in (**a**) and (**b**).

**Figure 2 viruses-15-00923-f002:**
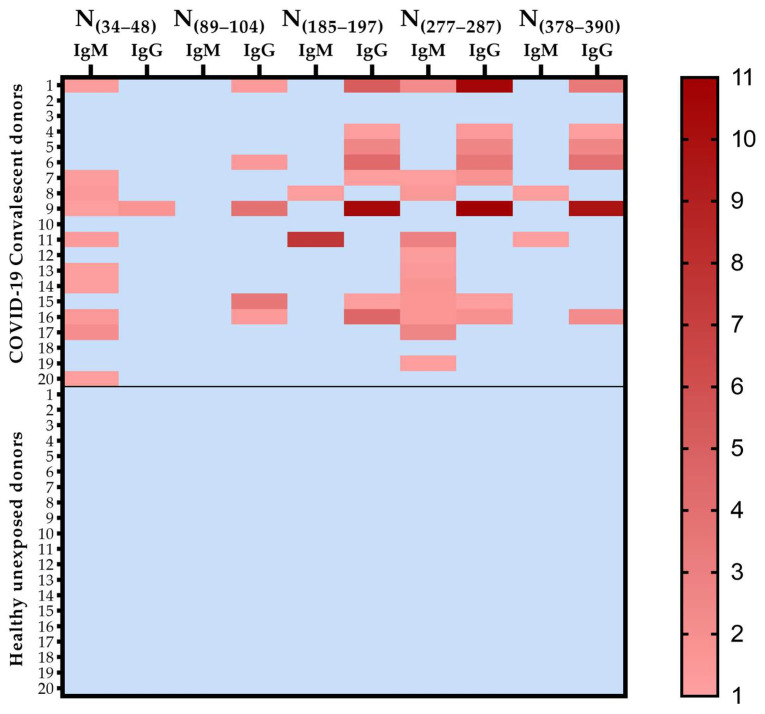
Heatmap of IgM and IgG reactivity indexes against synthetic epitopes. Values higher than 1 represent responder individuals and were indicated in the color scale, and non-responders were indicated by a light blue color.

**Figure 3 viruses-15-00923-f003:**
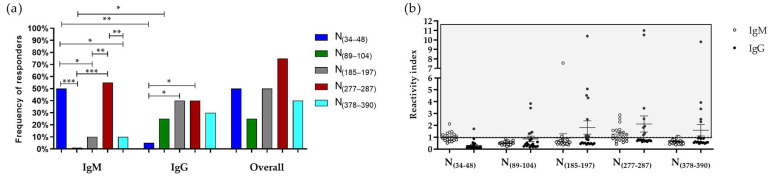
Evaluation of natural immunogenicity of predicted epitopes. (**a**) Frequencies of IgM, IgG, and overall responders to N_(34–48)_ (blue bar), N_(89–104)_ (green bar), N_(185–197)_ (gray bar), N_(277–287)_ (red bar), and N_(378–390)_ (light blue bar). The frequencies of responders to epitopes were compared by Fisher’s exact test, and statistical differences were indicated by asterisks: (*) = *p* < 0.05; (**) = *p* < 0.01; (***) = *p* < 0.001. (**b**) IgM (white dots) and IgG (black dots) reactivity indexes against predicted epitopes. Responders to peptides were indicated above the traced line.

**Figure 4 viruses-15-00923-f004:**
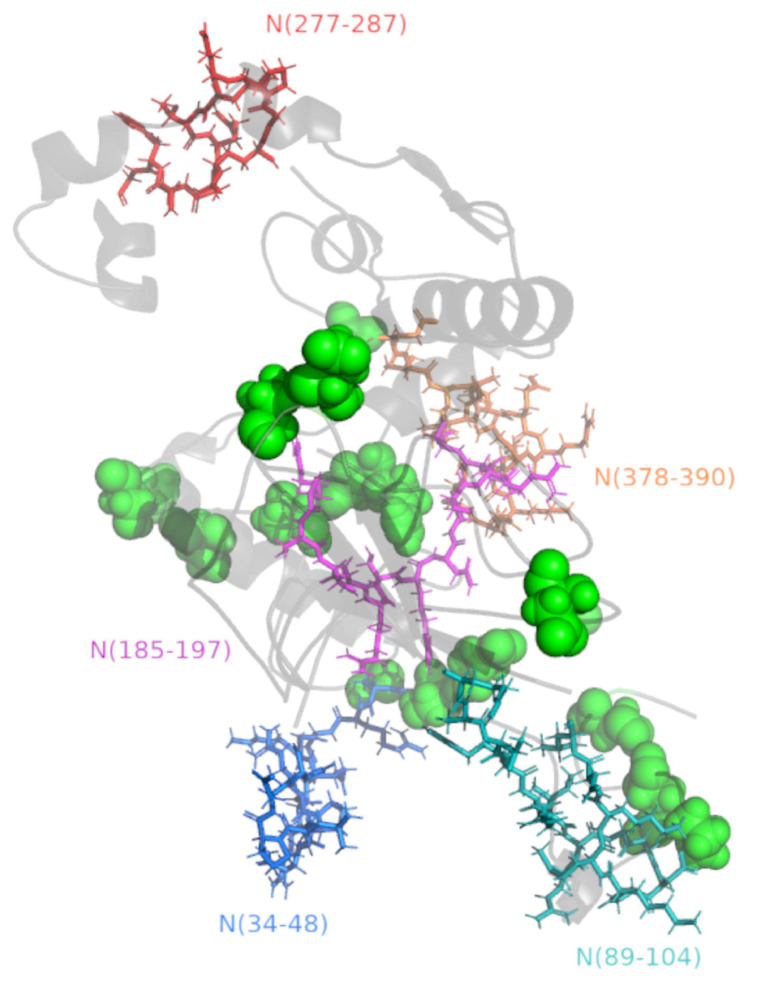
B-cell epitopes and key mutations in SARS-CoV-2 N protein—3D structure. The N protein was presented as cartoon (gray), in which the identified epitopes were represented by colored sticks: N_(34–48)_ (Blue), N_(89–104)_ (teal), N_(185–197)_ (magenta), N_(277–287)_ (red), and N_(378–390)_ (orange). The positions of key mutations (D3, Q9, P13, Del31/33, D63, P80, E136, R203, G204, T205, G215, L230, S235, D377, and S413) were represented by green spheres. In the cartoon, round helices, flat sheets, and smooth loops are applied to allow better visualization of the 3D structure.

**Table 1 viruses-15-00923-t001:** Evaluation of potential serological targets in SARS-CoV-2 N protein.

Epitope	Length	Sequence	Bepi Pred	ABC Pred	ESA	Ellipro	Vaxijen
N_(4–24)_	21	NGPQNQRNAPRITFGGPSDST	X	-	X	X	0.3231
**N_(34–48)_**	**15**	**GARSKQRRPQGLPNN**	X	X	X	X	**0.5955**
**N_(89–104)_**	**16**	**RATRRIRGGDGKMKDL**	X	X	X	X	**0.8755**
N_(115–127)_	13	TGPEAGLPYGANK	X	-	-	X	0.0561
**N_(185–197)_**	**13**	**RSSSRSRNSSRNS**	X	-	X	-	**10.062**
N_(254–264)_	11	ASKKPRQKRTA	X	-	X	-	0.2297
**N_(277–287)_**	**11**	**RGPEQTQGNFG**	X	-	X	X	**0.9248**
N_(323–331)_	9	EVTPSGTWL	X	-	-	X	0.4548
N_(363–376)_	14	FPPTEPKKDKKKKA	X	X	X	-	0.4801
**N_(378–390)_**	**13**	**ETQALPQRQKKQQ**	X	X	X	X	**0.8924**
N_(405–417)_	13	KQLQQSMSSADST	-	X	-	X	0.4364

The first column indicates the name of the epitope, representing the start and end position of the sequence. Vaxijen scores above 0.5 were considered antigenic. The “X” indicates that the algorithm in this column predicted, completely or partially, the sequence in the line, while the “-” indicates that the algorithm did not predict the sequence. Sequences predicted as antigenic linear B-cell epitopes were considered promising targets for serological tests and are highlighted in bold.

**Table 2 viruses-15-00923-t002:** Donors’ Characteristics.

Characteristics	COVID-19 Convalescent Donors (*n* = 20)	Healthy Unexposed Donors (*n* = 20)
Age (years)—median (IQR)	35.5 (30.75–40.25)	31 (20–38)
Gender	% (*n*)	
Male	40% (8)	35% (7)
Female	60% (12)	65% (13)
Diagnostic	% (n)	
RT-PCR	60% (12)	N/A
Serological test	30% (6)	N/A
RT-PCR and serological	10% (2)	N/A
Clinical aspects	median (IQR) or % (*n*)	
Symptomatic period (days)	12.5 (8–16)	N/A
Mild illness	80% (16)	N/A
Asymptomatic case	10% (2)	N/A
Hospitalization case	10% (2)	N/A
Symptoms	% (n)	
Fatigue	70% (14)	N/A
Fever	60% (12)	N/A
Headache	60% (12)	N/A
Cough	55% (11)	N/A
Diarrhea	40% (8)	N/A
Pharyngalgia	30% (6)	N/A
Coryza	30% (6)	N/A
Nausea	30% (6)	N/A
Dyspnea	25% (5)	N/A
Anosmia	10% (2)	N/A
Myalgia	5% (1)	N/A
Ageusia	5% (1)	N/A

Data were expressed as median (IQR) or % (n), where n is the total number of donors with available data. Healthy unexposed donors did not have data for diagnostic, clinical aspects, or symptoms, and because of that were signalized on the table by N/A.

**Table 3 viruses-15-00923-t003:** SARS-CoV-2 N-protein epitopes conservancy analysis among human coronaviruses.

	SARS	MERS	229E	NL63	OC43	HKU1
Epitope\Uniprot ID	P59595	K9N4V7	P15130	Q6Q1R8	P33469	Q5MQC6
N_(34–48)_	93.33%	33.33%	33.33%	26.67%	33.33%	33.33%
N_(89–104)_	87.50%	31.25%	37.50%	31.25%	37.50%	31.25%
N_(185–197)_	84.62%	76.92%	46.15%	61.54%	61.54%	53.85%
N_(277–287)_	100%	72.73%	45.45%	36.36%	45.45%	54.55%
N_(378–390)_	76.92%	38.46%	23.08%	30.77%	30.77%	30.77%

The values indicate the similarity between the SARS-CoV-2 epitope and the homolog protein in human coronaviruses (SARS-CoV, MERS-CoV, 229E, NL63, OC43, and HKU1). High (>80%), intermediate (80% > conservancy ≥ 50%), and low conservancy degrees (≤50%) were signalized by red, orange, and light blue, respectively. The proteins selected for the comparison between human coronaviruses were indicated by the Uniprot ID.

**Table 4 viruses-15-00923-t004:** Conservation of SARS-CoV-2 N protein amino acids recognized by nanobodies (7R98 and 7N0R) and the antibody 7CR5 across other human coronaviruses.

Organism	Antibody (PDB ID)	Residues Recognized by Antibodies	Aligned Residues in Other Human Coronaviruses					
			SARS	MERS	229E	NL63	OC43	HKU1
** *Lama glama* **	**7N0R**	N75	N76	N66	N47	G45	A89	A88
		T76	T77	A67	K46	K46	P90	F89
		N77	N78	N68	K47	G47	G91	G90
		D82	D83	Q73	53K	E51	E96	E95
		N153	N154	N142	E123	K121	S168	T167
	7R98	D81	D82	A72	N52	D50	T95	S94
		G137	G138	G127	G107	G105	Q152	Q151
		A138	A139	A128	A108	A106	A153	A152
		N140	N141	D130	T110	T108	V155	T154
*Homo sapiens*	7CR5	Q70	Q71	Q61	N42	N40	Q84	Q83
		Q160	Q161	Q149	N160	S128	R175	R174
		L161	L162	F150	Q131	I129	F176	F175
		P162	P163	A151	K132	A130	P177	P176
		T166	T167	K155	G136	E134	V181	I180
		K169	K170	K158	V139	V137	Q184	Q183

Non-conserved amino acids in human coronaviruses are indicated by gray cells.

## Data Availability

Data are available upon request for privacy or ethical reasons. The data presented in this study are available upon request from the corresponding author. The data are not publicly available due to confidential information related to donors’ personal data in accordance with the Institutional Ethics Committee of the Oswaldo Cruz Foundation.

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
