# Peer review of "Identification of B-Cell Linear Epitopes in the Nucleocapsid (N) Protein B-Cell Linear Epitopes Conserved among the Main SARS-CoV-2 Variants"

_viruses, 2023, doi:10.3390/v15040923_

Round 1

Reviewer 1 Report

Comments and Suggestions for Authors
The manuscript entitled “Identification of B-cell linear epitopes in the Nucleocapsid (N) protein B-cell linear epitopes conserved among the main SARS-CoV-2 variants.” by Dr. Rodrigues-da-Silva and colleagues identify computationally and validated experimentally B-cell linear epitopes in SARS-CoV-2 N protein, while their conservation across main SARS-CoV-2 variants and lineages was also evaluated. The work is interesting other than well conducted. The main limitation is the reduced sample size for experimental validations.  Please see below several suggestions for improving the manuscript which I consider suitable for publication in viruses MDPI:
a)    SARS-CoV-2 should be the correct annotation please revise the work, e.g., lines 39, 64 etc…
b)    Please include in lines 61-64 these two supporting references which describe in details the most recent diagnostic tests for SARS-CoV-2 detection PMID: 35744711 and PMID: 36950312
c)    Reference citation in 2.2 section should be uniformed according to the journal requirements of Viruses MDPI
d)    Please include at least mean ages and gender ratio of the study groups in section 2.5
e)    I suggest including a table reporting at least sensitivity, specificity, accuracy and negative and positive predictive values of the 11 selected peptides with the experiments conducted on 20 positive plus 20 negative donors’ sera
f)    For a better reading, I suggest moving the computational results reported in sections 3.4-3.6 above the experimental data reported in sections 3.2-3.3
g)    An important limitation is the reduced number of samples employed for the experimental validations. This point should be underlined in the discussion
h)    Line 383 I discourage mentioning figures/tables in the discussion
i)    Line 392 “didn't” should be did not. Please revise the manuscript for additional unconventional writing forms

Author Response

Dear Reviewer,

Firstly, we want to thank you for your time and efforts in helping us to improve the manuscript quality. We looking to attending and revising every comment and explaining here the modifications.

Best regards,

a)    SARS-CoV-2 should be the correct annotation please revise the work, e.g., lines 39, 64 etc…

We thank you for the observation. All annotations were revised as suggested.

b)    Please include in lines 61-64 these two supporting references which describe in details the most recent diagnostic tests for SARS-CoV-2 detection PMID: 35744711 and PMID: 36950312

We agree with the suggestion and include the mentioned references.

c)    Reference citation in 2.2 section should be uniformed according to the journal requirements of Viruses MDPI

Thank you for the observation. We revised the citations.

d)    Please include at least mean ages and gender ratio of the study groups in section 2.5

These data were presented in Table 2, Section 3.2. However, to comply with the suggestion, we have also included these data in Section 2.5, as shown below:

Samples from convalescent COVID-19 donors: Twenty individuals (12 women and 8 men), with ages ranging from 25 to 51 years (mean age: 35.8 ± 6.7 years) (…)

Healthy unexposed donors: 20 samples (13 women and 7 men), from blood donors, in Brazilian blood centers between the years 2010 and 2018 with ages ranging from 20 to 56 years (36.7 ± 10.4 years) (…)

e)    I suggest including a table reporting at least sensitivity, specificity, accuracy and negative and positive predictive values of the 11 selected peptides with the experiments conducted on 20 positive plus 20 negative donors’ sera.

We thank you for the suggestion, however, among eleven predicted epitopes, only five were also predicted as immunogenic (Vaxijen score>0.5) and tested against convalescent COVID-19 samples and samples from healthy individuals. The data of peptide recognition by samples were presented in section 3.3. Unfortunately, due to the limited number of individuals studied, we decided not to discuss the sensitivity, specificity, and accuracy of these epitopes. From our point of view, these characteristics need to be explored in a higher study. Therefore, we expected that the reviewer agrees with our option.

f)    For a better reading, I suggest moving the computational results reported in sections 3.4-3.6 above the experimental data reported in sections 3.2-3.3.

We understand the suggestion, however, we selected the order of results based on: (1) epitope prediction, (2) confirmation/evaluation of natural immunogenicity of epitopes, (3) evaluation of epitope conservation compared to SARS-CoV-2 variants and other human coronaviruses, and (4) evaluation of cross-reactivity of described antibodies to SARS-CoV-2 N protein. Therefore, although we agree with the proposed order of the sections, we believe that the order used leads to the expected understanding of our data.

g)    An important limitation is the reduced number of samples employed for the experimental validations. This point should be underlined in the discussion.

We appreciate the comment. Based on your observation, we have revised the discussion to emphasize this point, as shown below:

Notably, the number of individuals studied is a limitation of our study, but we believe it is sufficient to validate the natural immunogenicity of epitopes. In support of this point, the aforementioned study by Wang H. et al used samples from only 10 patients to identify peptides recognized by antibodies. However, we believe that a larger number of samples is needed to evaluate the true sensitivity and specificity of these epitopes in serological assays.”

h)    Line 383 I discourage mentioning figures/tables in the discussion.

We attended to the suggestion.

i)    Line 392 “didn't” should be did not. Please revise the manuscript for additional unconventional writing forms.

We appreciated the observation and revised the text.

Reviewer 2 Report

In this manuscript, authors identified five epitopes in the SARS-CoV-2 N protein, which were conserved in the main SARS-CoV-2 variants, by using immunoinformatic. These results support the effectivity of antigen tests for diagnosis of SARS-CoV-2. Overall, the quality of this manuscript was good, but the authors also made some rookie mistakes.

Suggestions for revision are as follows:

1.      In line 284, “(c)” should be replaced by “(b)”.

2.      In line 395, the word “line-ar” is wrong.

3.      In line 501, part of the information is lost.

4.      All the structural paragraphs have low resolution.

5.      You’ve mentioned the cold coronaviruses many times in this manuscript, so could you introduce this virus briefly?

Author Response

Dear Reviewer,

Firstly, we want to thank you for your time and efforts in helping us to improve the manuscript quality. We looking to attending and revising every comment and explaining here the modifications.

Best regards,

Suggestions for revision are as follows:

In line 284, “(c)” should be replaced by “(b)”.

We revised the text.

In line 395, the word “line-ar” is wrong.

Thank you for the observation. The text was revised.

In line 501, part of the information is lost.

Thank you again. We probably had a problem with the EndNote, but based on your observation we repaired this.

All the structural paragraphs have low resolution.

Sorry, but I don’t understand the term “structural paragraphs”. Based on this comment we revised all the text and figures, however, the figures had higher resolution if downloaded than when observed on text. We expected that the revision be the

You’ve mentioned the cold coronaviruses many times in this manuscript, so could you introduce this virus briefly?

We appreciate your suggestion and have improved the introduction as shown below:

“In addition to SARS-CoV-2 and its variants, there are four common cold coronaviruses (229E, NL63, OC43, and HKU1) that circulate in the human population, usually causing mild respiratory symptoms similar to the common cold, such as cough, runny nose, and sore throat, that should be investigated about cross-reactivity in COVID-19 diagnosis [9-12].”

Reviewer 3 Report

The manuscript of Rodrigues-da-Silva et al. describes immunoinformatics to identify five epitopes in the SARS-CoV-2 N protein (N(34-48), N(89-104), N(185-197), N(277-287), and N(378-390)) and validate their reactivity against samples from COVID-19 convalescent patients. The identified epitopes were fully conserved in the main SARS-CoV-2 variants, and highly with SARS-CoV, but low conserved with common cold coronaviruses (229E, 27 NL63, OC43, HKU1). The manuscript is very interesting and very well written. Despite the quality of the article, few points needs be addressed.

Minor review

Page 2, line 39:

Correct the virus name to SARS-CoV-2.

Line 41: “… as a pandemic disease [2].” Please remove the word “disease”. 

Line 44: “the spread of the disease”. Please, change to “the spread of the virus.” 

Finally, I was wondering if the authors could do a comment about the useful of these N protein as vaccine target once there is conserved epitopes. 

Finally, maybe the authors could comment about the useful of the N protein as vaccine target once it was reported these conserved epitopes? Of course, the antibody response would not be neutralizing, but perhaps, these antibodies could be able to activate complement system or trigger a cytotoxicity mediated by cell dependent of antibody. I would like to hear the opinion of the authors.   

Author Response

Dear Reviewer,

Firstly, we want to thank you for your time and efforts in helping us to improve the manuscript quality. We looking to attending and revising every comment and explaining here the modifications.

Best regards,

Correct the virus name to SARS-CoV-2.

Thank you for the observation. The virus name was revised in all text.

Line 41: “… as a pandemic disease [2].” Please remove the word “disease”. 

The text was revised as suggested.

Line 44: “the spread of the disease”. Please, change to “the spread of the virus.” 

We agree with the suggestion and change the text.

Finally, I was wondering if the authors could do a comment about the useful of these N protein as vaccine target once there are conserved epitopes.  

Finally, maybe the authors could comment about the useful of the N protein as vaccine target once it was reported these conserved epitopes? Of course, the antibody response would not be neutralizing, but perhaps, these antibodies could be able to activate complement system or trigger a cytotoxicity mediated by cell dependent of antibody. I would like to hear the opinion of the authors.   

We appreciate this suggestion and thank you. Although antibodies against N protein are not neutralizing, this protein has been considered a vaccine candidate, with the spike protein, based on its role in inflammation, cell death, antiviral innate immunity, and antiviral adaptive immunity [1,2]. However, the use of N protein in vaccine design is mainly associated with cellular immune response, which should be pivotal for protection against infection by SARS-CoV-2 variants of concern, based on the high conservation of the full-length N-protein [3]. In this context, despite the interest in evaluating the potential of this antigen as a vaccine candidate, from our point of view, this type of discussion could interfere with a better understanding of our data since we did not attempt to evaluate T-cell epitopes in this work. Moreover, despite the conservation of the N protein among human coronaviruses, the proteins from HCoV-HKU1, SARS-CoV, and MERS-CoV do not appear to play the same role in endothelial cells as the SARS-CoV-2 N protein [4].

In addition, due to the preclinical development of an mRNA vaccine in my laboratory using N protein and S protein, we prefer to leave this discussion of the potential of N protein as a vaccine candidate for the future. Finally, thank you again for the comment and I apologize for not including this discussion in the manuscript.

PS: I list below the mentioned references to embased this opinion.

  1. Hajnik, R.L.; Plante, J.A.; Liang, Y.; Alameh, M.G.; Tang, J.; Bonam, S.R.; Zhong, C.; Adam, A.; Scharton, D.; Rafael, G.H.; et al. Dual spike and nucleocapsid mRNA vaccination confer protection against SARS-CoV-2 Omicron and Delta variants in preclinical models. Science translational medicine 2022, 14, eabq1945, doi:10.1126/scitranslmed.abq1945.
  2. Chiuppesi, F.; Nguyen, V.H.; Park, Y.; Contreras, H.; Karpinski, V.; Faircloth, K.; Nguyen, J.; Kha, M.; Johnson, D.; Martinez, J.; et al. Synthetic multiantigen MVA vaccine COH04S1 protects against SARS-CoV-2 in Syrian hamsters and non-human primates. NPJ vaccines 2022, 7, 7, doi:10.1038/s41541-022-00436-6.
  3. Chiuppesi, F.; Zaia, J.A.; Faircloth, K.; Johnson, D.; Ly, M.; Karpinski, V.; La Rosa, C.; Drake, J.; Marcia, J.; Acosta, A.M.; et al. Vaccine-induced spike- and nucleocapsid-specific cellular responses maintain potent cross-reactivity to SARS-CoV-2 Delta and Omicron variants. iScience 2022, 25, 104745, doi:10.1016/j.isci.2022.104745.
  4. Qian, Y.; Lei, T.; Patel, P.S.; Lee, C.H.; Monaghan-Nichols, P.; Xin, H.B.; Qiu, J.; Fu, M. Direct Activation of Endothelial Cells by SARS-CoV-2 Nucleocapsid Protein Is Blocked by Simvastatin. J Virol 2021, 95, e0139621, doi:10.1128/JVI.01396-21.
